# Ranking of Assets with Respect to Their Exposure to the Landslide Hazard: A GIS Proposal

**Paolino Di Felice** 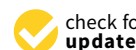

Department of Industrial and Information Engineering and Economics, University of L'Aquila,
67100 L'Aquila, Italy; paolino.difelice@univaq.it; Tel.: +39-320-423-2040

**Abstract:** The need to protect critical infrastructures (for short called assets within this paper) arises because of the hazards they are exposed to. In this article, the hazard is represented by the landslides. The first part of the paper proposes a scientifically robust method for the identification of the top-N assets that can be modeled as "points" (mainly buildings). The developed method takes into account the slope of the terrain, the runout distance of the landslide and its trajectory. The latter is roughly estimated through the notion of linear regression line. The method is applied to a real case to carry out a preliminary validation of it. In the second part of the paper, it is formalized the problem of computing the ranking of assets that can be modeled as "lines" (e.g., highways, power lines, pipelines, railway lines, and so on, that cross a given territory). The problem is solved in three steps: (a) Segmentation (it "cuts" each route in segments), (b) Sampling (it extracts points from each segment), and (c) Calculation (it associates an exposure value to each extracted point and, then, computes the exposure of the various segments composing the routes). The computation of the exposure for the points is carried out by applying the method of the first part of the paper. Both rankings can be used by the local administrators as a conceptual tool for narrowing down a global problem to smaller, higher exposure, geographic areas where the management of the hazard is crucial.

**Keywords:** hazard; buildings; road lines; railway lines; pipelines; landslides; exposure; ranking; GIS

## 1. Introduction

In recent years, many nations worldwide have recognised the increasing importance to protect their key infrastructures (such as power plants, bridges, highways, airports, and so on). For example, in 2008, the European Council issued Directive 2008/114/EC, which required the Member States to identify European critical infrastructures and assess the needs for their protection. This Directive defined "critical infrastructure" as: an asset, system or part thereof located in Member States which is essential for the maintenance of vital societal functions, health, safety, security, economic or social well-being of people, and the disruption or destruction of which would have a significant impact in a Member State as a result of the failure to maintain those functions. Setola et al. [1] provide an up-to-date report of the initiatives of several governments about the protection and resilience of their critical infrastructures.

The protection of the States' critical infrastructures is the focus of the present paper, where they are called assets. The need to protect the assets arises because of the multiple hazards (e.g., earthquakes, floods, landslides) they are exposed [2]. In the present study, the attention is focused on landslides which play a very important role in many countries of the world [3]. In Italy landslides are frequent and cause considerable damage and many casualties each year [4,5]. According to the outcome of a

study carried out by Jaedicke et al. [6], "Italy has the highest number of people exposed to landslide hazard among the European countries."

Landslides become a hazard when they interact with the so-called "elements at risk". The evaluation of the side effects caused by landslides is usually referred to as risk assessment. In mathematical terms, the risk can be computed as in [6]:

$$Risk = Hazard \times Vulnerability \times ElementsAtRisk \tag{1}$$

*Hazard = Susceptibility × Trigger*. Vegetation cover, soil moisture, slope, and lithology are usual susceptibility factors, while precipitation is the prominent triggering factor. Vulnerability denotes the extent of the damage to an element at risk, expressed in a scale from 0 (no damage) to 1 (total destruction). The quantification of the Hazard parameter induced by landslides is a complex task; reference [6] proposes a method for its determination.

Many publications are available about the assessment of the vulnerability of elements at risk (for example, [7–9]), as well as on the assessment of the risk caused by landslides (for example, [6,10–12]). In all the proposals, the identification of the elements at risk of landslides is carried out by computing the intersection of the geometries modeling the assets and the hazard area (e.g., [13]). The consequence is that the elements in the region of study are split into two disjoint categories: one containing the elements that are exposed to the landslide hazard, while the other contains the remaining ones. Notice that the elements in the former category are all "ex-aequo" with respect to the "level of exposure" to the landslide hazard.

When the number of assets to be monitored is large (for example, in Italy there are 72,355 schools located in 43,643 separate buildings, while the total number of public buildings is far greater), returning to the practitioners in risk mitigation the ranking of the assets they are in charge of guides them in prioritizing the controls on the field. In fact, in that scenario they may limit the detailed risk assessment to the assets with a value of the exposure above a given threshold (briefly the top-N assets). This way the overall processing time required for the computation of the vulnerability, and hence of the risk, is reduced dramatically.

The present study focuses on assets modeled both as points (i.e., buildings, of any kind), and as lines (e.g., roads, highways, railway lines, pipelines, power lines). To a preliminary and, necessarily, high level of abstraction, the problems that are studied can be described as follows. Given a reference territory (for example an Italian region) and:

- The set of buildings of a given category (for example the railway stations), return their ranking with respect to the level of exposure to the landslide hazard;
- The set of lines (for example the railway lines) that cross it, return, for each of them, the ranking of their stretches with respect to the level of exposure to the landslide hazard.

In reference [14], the authors studied the landslide exposure of buildings. Starting from vector data about the elements at risk, they derive raster data. The latter are used to output a multilayer-exposure map about potential hotspots for an in-depth analysis of vulnerability and consequent risk. The disadvantage of this method is that a ranking of hotspots is not possible. The ranking problem has origin in the Information Retrieval domain, while methods for the ranking of buildings have been proposed much more recently. In [15], for example, a method has been proposed to construct the ranking of buildings with respect to the fire hazard, while [16] formalizes a method to rank the illegal buildings located close to rivers. Another method for ranking the buildings with respect to their environmental performance can be found in [17]. To the best of our knowledge, Refs. [18,19] are the first attempt to introduce a method for ranking the buildings present in a large territory, with respect to the level of exposure to the landslide hazard.

For both the problems studied, the following are given: (a) a theoretical solution, (b) its implementation with open source GIS software, (c) experimentation through case studies, (d) a

preliminary validation of the proposed methods and, finally, (e) the discussion of the results. The formalization of the two novel methods is the actual contribution of the present paper.

The manuscript is structured in two parts, each is a paper in itself. The first one (Sections 2–4) focuses on assets modeled as points, while the remaining pages concern assets modeled as lines. The sections about assets modeled as points extend a previous paper [20], by adding the formalization of the method to compute the ranking and its validation.

In detail, Section 2.1 concerns the formalization of the definitions and notations on which is based the method for calculating the ranking of buildings, located in the reference territory, with respect to the level of the potential hazard of being hit by landslides. Section 2.2 proposes the equations for calculating the level of exposure to the landslide hazard of buildings located within the study area. Section 2.3 touches on the way the theory was implemented. The core component of the solution is the DataBase Management System equipped with a Spatial extension (briefly Spatial DBMS). Moreover, the section lists the tables of the Spatial DataBase (briefly SpatialDB) and the names of the User Defined Functions that implement the proposed method.

Section 3 presents a case study about the 114 railway stations of the Abruzzo region (Italy).

Section 4 concerns the validation of the results of the case study and, hence, a preliminary evaluation of the method.

Section 5 focuses on the problem of calculating the ranking, with respect to the landslide hazard, of assets modeled through the geometric primitive "line".

The case study adopted for the experimentation of the proposed method (Section 6) concerned the nine railway lines crossing the Abruzzo region.

Section 7 closes the paper by making a balance between the initial objectives and the results.

## 2. Materials and Methods for Assets Modeled as Points

### 2.1. Definitions and Notations

Hereafter are listed the definitions and notations on which is based the method for ranking this category of assets.

**Definition 1.** *GeoArea. This term denotes the reference territory. The GeoArea can be a region or a state. GeoArea is defined as the tuple $\langle ID, description, boundary \rangle$, where ID is an identifying code, description is a string that specifies the name of the reference territory and, finally, boundary denotes the geometry of the border of the aforementioned territory.*

**Definition 2.** $\mathcal{Z}$ *(Zones)* $= \{z_k(k = 1, 2, ...) \mid z_k$ *is a zone in the GeoArea}. card($\mathcal{Z}$) denotes the cardinality of set $\mathcal{Z}$. The elements in $\mathcal{Z}$ make a full partition of the GeoArea. The generic zone (i.e., $z_k$) can be modelled as a "simple polygon", as meant in the OpenGIS Abstract Specification [21]. Each element of $\mathcal{Z}$ is defined by the tuple $\langle ID, boundary of z_k, Sz_k \rangle$, being ID an identifying code. $Sz_k$ is a numerical value that quantifies the (spatial) probability that $z_k$ produces landslides. The value of $Sz_k$ ranges from 0 to 1. Brabb [3], introduced the term susceptibility to denote such a quantitative estimate.*

Assessing and mapping the landslide susceptibility of a GeoArea have been a relevant issue, e.g., [22–26]. In the paper, $z_k$ is an overloaded notation since it represents the *ID* of a zone and its *geometry* as well. The context suggests the correct interpretation. Our zones are the *slope units* in [26], since they are the basic units for assessing landslides. This is the fundamental assumption of the present paper. Modelling the GeoArea as a partition of slope units ensures that the evaluation results are more adherent to the reality. This result is not guaranteed by adopting as map units, for instance, the *grid cells*. In this latter choice, in fact, the GeoArea is split into regular grids; the limit of this partitioning method is that it destroys the integrity of the slopes. The greater the slope gradient is, the bigger is the value of $Sz_k$. Set $\mathcal{Z}$ is an input dataset of the two problems studied in this paper.

Therefore, the discussion of the construction algorithm of the slope units that partitions the GeoArea is out of scope. Reference [26] is a recent reference on the topic.

**Definition 3.** $\mathcal{B}$ *(Buildings)= $\{b_i(i = 1, 2, ...) \mid b_i$ is a building inside the boundary of the GeoArea}. In the following, the subscript i is used always as a shorthand for the element $b_i$ of set $\mathcal{B}$. $card(\mathcal{B})$ denotes the cardinality of set $\mathcal{B}$. Each building in $\mathcal{B}$ is defined by the tuple $\langle ID, description, geom, exposure \rangle$. ID uniquely identifies the building; description is a string that describes the building; geom is the footprint of $b_i$ in some reference system; finally, exposure is a positive numeric value denoting the level of (spatial) exposure of $b_i$ to the landslide hazard.*

**Definition 4.** *$HazardArea_i$. The method we are going to introduce for the computation of the value of the exposure of buildings to the landslides is based on the conjecture that only the surroundings zones might pose a threat to them. This conjecture is inspired by the Tobler's first law of geography [27]: "Everything is related to everything else, but near things are more related than distant things." Within this paper, a circular region of radius r, centered on the geometric centroid of $b_i$, formalizes this idea. We call it $HazardArea_i$ (Figure 1). An abstract r can be infinite, but for the problem at hand its value is limited.*

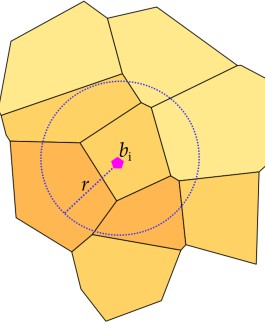

**Figure 1.** The $HazardArea_i$ around building $b_i$ (whose footprint is denoted as a fuchsia pentagon). A more intense color of the *zone* corresponds to a greater value of its $Sz_k$.

**Definition 5.** *EXP (Exposure) = $\{exp_i(i = 1, ..., card(\mathcal{B})) \mid exp_i$ is the value of the exposure of building $b_i$}. Set EXP collects the values of the exposure of all the buildings in $\mathcal{B}$.*

**Definition 6.** *$\mathcal{NZ}$ (Nearest Zones) = $\{nz_{i,j}(i = 1, ..., card(\mathcal{B})), (j = 1, ..., card(\mathcal{NZ})) \mid nz_{i,j} \in \mathcal{Z} \cap HazardArea_i$ is a zone located inside $HazardArea_i$}. $nz_{i,j}$ can represent either a whole zone or the portion of it result of the intersection with the $HazardArea_i$ (Figure 2a). $nz_{i,j}$ is an overloaded notation since it rapresents both the ID of a nearest zone and its geometry.*

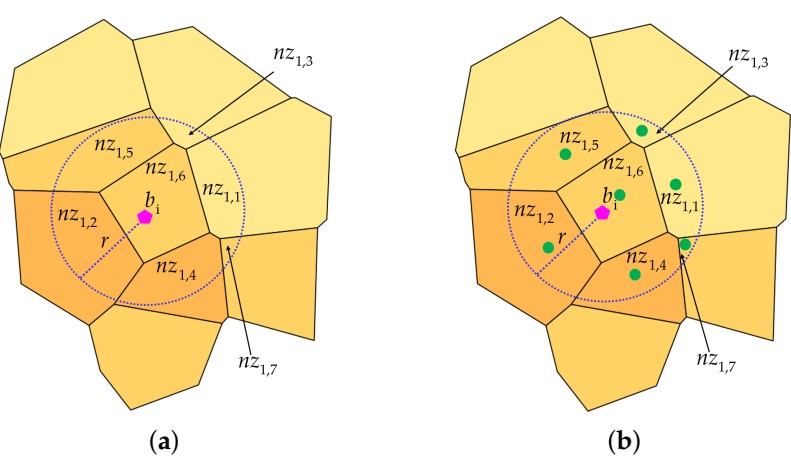

(**a**)                         (**b**)

**Figure 2.** The nearest zones ($nz_{i,j}$) inside the $HazardArea_i$ about building $b_i$ (**a**) and their centroids (the green dots) (**b**); ($i = 1, j = 1.7$).

**Definition 7.** $cnz_{i,j}$ (Centroid of $nz_{i,j}$) denotes the geometric centroid of $nz_{i,j}$. The ground position of this point is described by its geographical coordinates. Figure 2b shows, with green dots, the centroids of the $nz_{1,j}$ of Figure 2a.

**Definition 8.** $\mathcal{CL} = \{cl_p (p = 1, 2, ...) \mid cl_p$ is a contour line of the GeoArea}. A contour line is a curve that joins points at the same altitude with respect to the sea level. $cl_p$ is described by the tuple $\langle ID, elevation, geometry \rangle$. ID uniquely identifies the curve, elevation the relative elevation and, geometry its shape.

**Definition 9.** $\mathcal{NCL}$ (Nearest Contour Lines) $= \{ncl_{i,o} (i = 1, ..., card(\mathcal{B})), (o = 1, ..., card(\mathcal{CL})) \mid ncl_{i,o} \in \mathcal{CL} \cap HazardArea_i$ is a nearest contour line inside Hazard Area_i}. $ncl_{i,o}$ denotes either an entire contour line or portions of it (as the result of the intersection with the $HazardArea_i$). $ncl_{i,o}$ is an overloaded notation since it raprresents both the ID of a nearest contour line and its geometry as well. Figure 3 proposes the four nearest contour lines $ncl_{i,o}$.

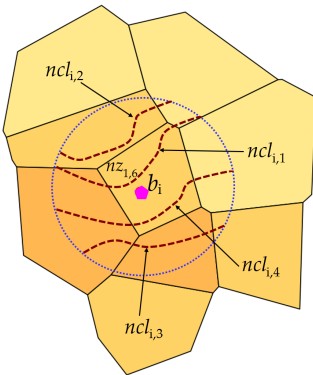

**Figure 3.** The nearest contour lines ($ncl_{i,o}$; $o = 1.4$—the dotted lines) to building $b_i$.

**Definition 10.** $\mathcal{ZF}$ (Zone Fragments) $= \{zf_{i,j,t} (i = 1, .., card(\mathcal{B}), j = 1, ..., card(\mathcal{NZ}), (t = 1, ...)) \mid zf_{i,j,t}$ is a zone fragment inside the nearest zone $nz_{i,j}$ obtained by splitting $nz_{i,j}$ with the nearest contour lines that cross it}. $\mathcal{ZF}$ is the set of the zone fragments produced by splitting the $nz_{i,j} \in \mathcal{NZ}$ against the $ncl_{i,o} \in \mathcal{CL}$. Figure 4a shows the three zone fragments in which $nz_{1,6}$ is fragmented by the split operation against the two contour lines crossing it. Visually, those zone fragments are the areas delimited by $nz_{1,6}$ and the portion of the nearest contour lines that cross it.

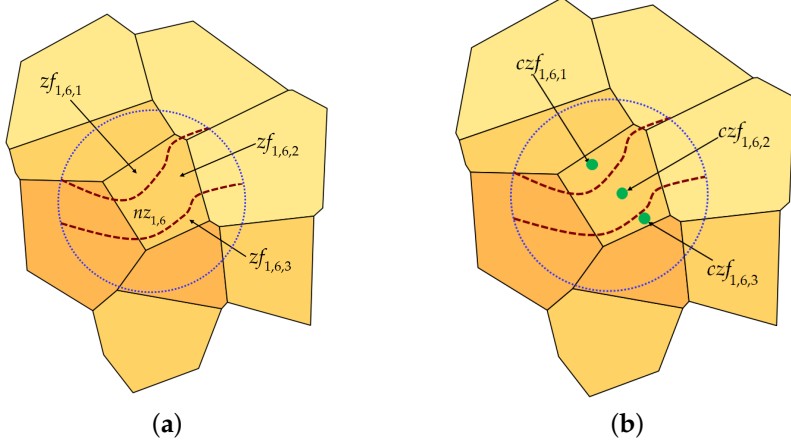

(**a**) 　　　　　　　　　　　　　　　　(**b**)

**Figure 4.** The zone fragments ($zf_{1,6,t}$) of the nearest zone $nz_{1,6}$ (**a**), and their centroids ($czf_{1,6,t}$) (**b**); $t = 1.3$.

**Definition 11.** $czf_{i,j,t}$ (Centroid of $zf_{i,j,t}$) denotes the geometric centroid of $zf_{i,j,t}$ whose position is described by its geographical coordinates, denoted as: $(czfx_{i,j,t}, czfy_{i,j,t})$. Figure 4b shows the three centroids (the green dots) of the zone fragments $zf_{1,6,t}$ of Figure 4a.

**Definition 12.** $\mathcal{LR}$ (Linear Regressions) = $\{lr_{i,j}(i = 1, ..., card(\mathcal{B})), (j = 1, ..., card(\mathcal{NZ})) \mid lr_{i,j}$ denotes the linear regression line of the nearest zone $nz_{i,j}$. $lr_{i,j}$ passes through the centroid $(cnz_{i,j})$ of $nz_{i,j}$ and has angular coefficient m such that, given the t centroids $(czfx_{i,j,t}, czfy_{i,j,t})$ of the zone fragments $zf_{i,j,t}$, the sum of their squared residuals is minimal. The line $lr_{i,j}$ is described by Equation (2), while Equations (3) and (4) specify the condition about the sum of the squares of the residuals.

$$y - cnzy_{i,j} = m(x - cnzx_{i,j}) \tag{2}$$

$$S(m,q) = \sum_{i=0}^{n} (m * czfx_{j,t} + q - czfy_{j,t})^2 \tag{3}$$

$$\text{where } q = cnzy_{i,j} - m * cnzx_{i,j} \tag{4}$$

The linear regression line is used to estimate the direction of fall of the landslides caused by the zones located near the assets.

**Definition 13.** $\mathcal{BLR}$ (Buffered Linear Regressions) = $\{blr_{i,j}$ $(i = 1, ..., card(\mathcal{B})), (j = 1, ..., card(\mathcal{NZ})) \mid \forall lr_{i,j} \in \mathcal{LR}, blr_{i,j}$ is the geometry returned by the operation of line buffering of $lr_{i,j}\}$. $blr_{i,j}$ models the wake of the landslide from $nz_{i,j}$, in case such an unfortunate event takes place. The outcome of the buffering operation is the construction of a "corridor" (of side $2 \times l$) around the linear regression line $lr_{i,j}$ (Figure 5).

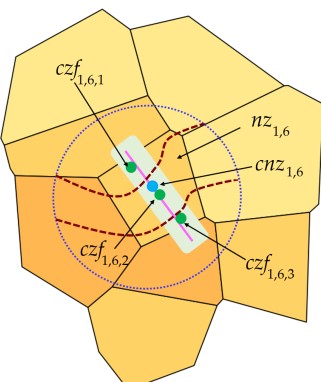

**Figure 5.** Display of the output of the operation of *line buffering* of $lr_{i,j}$. $cnz_{1,6}$ (the blue dot) is the centroid of the nearest zone ($nz_{1,6}$) where the line runs through.

**Definition 14.** $\mathcal{LS}$ (LandSlides) = $\{ls_{i,j}(i = 1, ..., card(\mathcal{B})), (j = 1, ..., card(\mathcal{LS}))\}$ with $\mathcal{LS} \subseteq \mathcal{BLR}$, such that $\mathcal{LS} = (\mathcal{BLR} \cap geom\_of\_b_i) \neq \emptyset$. The elements of set $\mathcal{LS}$ are the only $blr_{i,j}$ that intersect the geometry of building $b_i$, that is, the only landslides that can hit $b_i$ in case of a triggering event. The red corridor in Figure 6 depicts a buffered linear regression that impacts on the (only partially visible) building; while the green corridor does not constitute a threat to the same building.

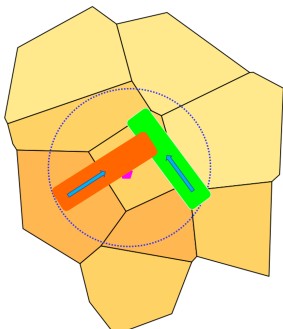

**Figure 6.** Two examples of *Buffered Linear Regressions*. The red one is a *landslide*. The arrows show the direction of the movement of the landslide, in case of activation.

**Definition 15.** $\mathcal{LSZ}$ *(LandSlide Zones)* $= \{lsz_{i,j}(i = 1, ..., card(\mathcal{B})), (j = 1, ..., card(\mathcal{LSZ}))\}$ *with* $\mathcal{LSZ} \subseteq \mathcal{NZ}$ *such that* $\mathcal{LSZ}$ *is the set of* $nz_{i,j}$ *that can generate landslides able, potentially, to hit building* $b_i$. $lsz_{i,j}$ *denotes the zone from which originates the landslide* $ls_{i,j}$. *Since* $\mathcal{LSZ}$ *is a subset of set* $\mathcal{NZ}$, *it follows that the element* $lsz_{i,j}$ *is described by the same tuple* $\langle ID, boundary, Sz_{i,j} \rangle$ *of* $nz_{i,j}$.

### 2.2. The Ranking Method

The problem to be solved is the following:

*given a reference territory (the GeoArea—for example an Italian region) and the set of assets that can be modeled as points placed in it (the set* $\mathcal{B}$*—for example railway stations), calculate their ranking with respect to the level of exposure to the landslide hazard.*

In general, the exposure is evaluated in relation to a potential scenario regardless of the probability of occurrence of the natural hazard taken into account [2]. In this paper, the exposure is evaluated considering a scenario defined by the trajectory of potential landslides.

In detail, the developed method takes into account the zone gradient, the runout distance of the landslide [10], and its trajectory. The gradient is captured through $Sz_k$ (see Definition 2). The runout distance of the landslide is implemented by limiting the calculation of the level of exposure of buildings by examining only the zones that intersect the $HazardArea_i$ (Definition 4). Finally, the trajectory of the landslides is captured through the notion of linear regression line (Definition 12). The landslides' trajectory is computed by taking into account the contour lines that intersect the $HazardArea_i$, that is the Nearest Contour Lines ($\mathcal{NCL}$, Definition 9). The latter allow to further subdivide the zones that intersect the $HazardArea_i$ into fragments whose centroids are the basis of the regression line computation. Once obtained the straight line ($lr_{i,j}$) that estimates the landslide trajectory of the nearest zone $nz_{i,j}$, it remains to be determined whether it may hit building $b_i$.

**Definition 16.** $IIls_{i,j}$ *(Impact Index) is a decimal value in the interval* $(0, 1]$ *defined as the ratio between the area of the geometry result of the operation* $(geom\_of\_b_i \cap ls_{i,j})$ *and the area of* $geom\_of\_b_i$, *(Equation (5))*. $IIls_{i,j} = 1$ *if the landslide corridor covers the whole geometry of building* $b_i$. *The definition of set* $\mathcal{LS}$ *implies that* $IIls_{i,j} > 0$.

$$IIls_{i,j} = \frac{Area(geom\_of\_b_i \cap ls_{i,j})}{Area(geom\_of\_b_i)} \tag{5}$$

*Figure 7 shows the two types of impact of a landslide on the building* $b_i$. *The impact index* $IIls_{i,j}$ *has not to be confused with the magnitude of the impact (if any) of the landslide* $ls_{i,j}$ *against building* $b_i$ *that we are not able to compute since the latter depends upon the speed of the landslide material and the building construction material, factors outside the scope of the present paper.*

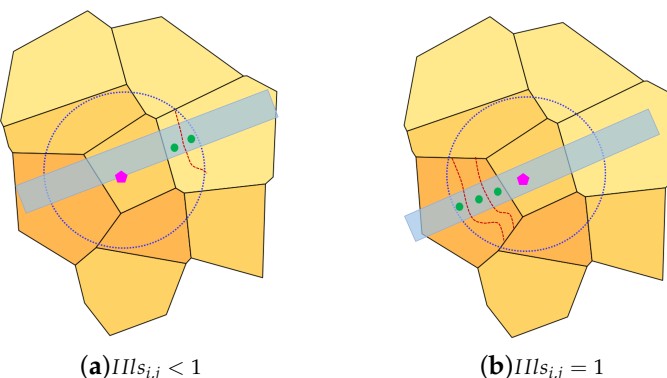

**Figure 7.** Display of two types of impact of a landslide on a building.

The level of potential exposure of building $b_i$ to the landslide hazard is equal to the sum of the partial exposure values $p\_exp_{i,j}$ (Definition 17) attributable to each of the (seenearest zones whose (seelandslides $ls_{i,j}$ potentially may hit $b_i$.

**Definition 17.** $p\_exp_{i,j}$ *(Partial Exposure)*, $(i = 1, ..., card(\mathcal{B})), (j = 1, ..., card(\mathcal{LSZ}))$, *is the value of the partial exposure of building $b_i$ to the landslide $ls_{i,j}$.*

Equation (6) computes the value of the partial exposure, where $Area_{lsz_{i,j}}$, $Sz_{i,j}$, and $IIls_{i,j}$ denote, in order, the extension of the area of the landslide zone $lsz_{i,j}$, the susceptibility value of $lsz_{i,j}$, and the value of the impact index.

$$p\_exp_{i,j} = Area_{lsz_{i,j}} \times Sz_{i,j} \times IIls_{i,j} \tag{6}$$

Equation (7) computes the total value of the level of exposure of building $b_i$, where $n = Card(\mathcal{LSZ})$.

$$exp_i = \sum_{j=1}^{n} p\_exp_{i,j} \tag{7}$$

Algorithm 1 (*ComputeBuildingExposure*) lists the steps for computing the exposure of a generic building ($b_i$).

---

**Algorithm 1:** ComputeBuildingExposure

**Data:** $GeoArea$, $\mathcal{Z}$, $b_i$, $\mathcal{CL}$, $r$, $d$
**Result:** $exp_i$
1   $HA_i \leftarrow$ Compute a buffer of radious $r$ around $b_i$;
2   $\mathcal{NZ} \leftarrow$ Compute the Nearest Zones set;
3   $\mathcal{NCL} \leftarrow$ Compute the Nearest Contour Lines set;
4   $exp_i \leftarrow 0$;
5   **foreach** $z_j$ *in* $\mathcal{NZ}$ **do**
6      $p\_exp_{i,j} \leftarrow 0$;
7      $\mathcal{ZF} \leftarrow$ Compute the set of fragments for zone $z_j$;
8      $\mathcal{BLR} \leftarrow$ Compute the buffered linear regression line;
9      **if** *($\mathcal{BLR}$ intersects geom_of_$b_i$)* **then**
10        $p\_exp_{i,j} = Area_{lsz_{i,j}} \times Sz_{i,j} \times IIls_{i,j}$
11      **end**
12      $exp_i = exp_i + p\_exp_{i,j}$;
13   **end**
14   **return** $exp_i$;

---

The formalization of the problem and the solution proposed in this paper are a consolidation of what is proposed in [18,19]; consolidation achieved by taking into account the altimetric evolution of the reference territory.

## 2.3. Spatial DataBase and Implementation of the Method

PostgreSQL/PostGIS is the software technology we have chosen for carrying out the implementation of the developed theory. Reference [28] lists the multiple advantages that the Spatial DBMSs technology offers in the processing of geographical data.

The design of the SpatialDB was articulated in the usual two steps: Conceptual Design and Logical Design. The four tables of the SpatialDB are shown below. The underlined attribute denotes the primary key of the table it belongs to.

- `GeoArea(`<u>`id`</u>`, geom);`
- `Zones(`<u>`id`</u>`, Szk, geom);`
- `ContourLines(`<u>`id`</u>`, elevation, geom);`
- `Buildings(`<u>`id`</u>`, name, geom, exposure).`

The implementation of the theory took place using the PL/pgSQL language of PostgreSQL for writing User Defined Functions. Six UDFs have been written: `_NearestZonesFinder();` `_NearestContourLinesFinder(); _ZoneFrangmentFinder(); _BufferedLinearregressionFinder();` `_LandSlideFinder();` and `_ContributionOfLandSlide().` Their execution in sequence returns the value of the exposure parameter for the current building and its storage into the `exposure` column of the `Buildings` table of the SpatialDB.

## 3. Case Study: The Railway Stations of Abruzzo

In [19], Di Felice et al. have experimented their ranking method on the public schools in the Abruzzo region. In the present work it was decided to stay within the borders of Abruzzo, but to refer to another category of public buildings: the railway stations. To achieve an effective and complete presentation of the results by means of maps, we put a GIS software beside PostgreSQL/PostGIS. The choice fell on QuantumGIS. It guarantees full compatibility with PostGIS.

### 3.1. The Input Data

GeoArea. It coincides with the boundary of the Abruzzo region; an area of 10,800 km$^2$ and 1,330,000 inhabitants.

Set $\mathcal{B}$. Figure 8 shows the map about the 114 railway stations in the Abruzzo region.

$\mathcal{CL}$. The shapefile of the contour lines of the Abruzzo territory have been generated (through QGIS) from a raster data of such a region. Then, the contour lines (returned by QGIS) have been intersected with the geometry of the boundary of the Abruzzo region (the GeoArea).

$\mathcal{Z}$. The set $\mathcal{Z}$ comes from a shapefile courtesy of the Abruzzo region. Below, we summarize what we have learnt by reading the metadata about such a shapefile. It is composed of 22,121 records (our zones) having an average area of 0.49 km$^2$. In the scholarly jargon, this shapefile represents the landslide susceptibility of the Abruzzo region. It was carried out by implementing the steps of the statistical classification method proposed by Guzzetti et al. [23] and applied to prepare the landslide susceptibility for the Collazzone study area (in the Umbria region of Italy). The data used to produce the shapefile concerned the landslide inventory of the Abruzzo (a database about the distribution and characteristics of 3236 landslides occurred in the area from 1946 to 2018) and a set of landslide influencing factors. The categories of causative factors are: geological (e.g., lithology), geomorphological (e.g., slope degree, slope length, altitude), environmental (e.g., land use), and hydrological (e.g., distance from rivers).

The shapefile is delivered with the value of three different metrics measuring its ability to classify the landaslides collected in the inventory database. The metrics are the Cohen's Kappa index, the True

Positives and the True Negatives. Their values are, in order: 0.66 (range: $(-\infty, 1]$), 0.76 (range: $[0, 1]$) and 0.81 (range: $[0, 1]$). These values are similar to those obtained by Guzzetti et al. with regard to the susceptibility model they proposed and adopted to produce the susceptibility map of the Collazzone area [23]. References [25,29] report that frequency ratio methods are able to produce better landslide susceptibility files; but at the time of writing, this kind of shapefile was not available for the Abruzzo region.

The value of *r* can be set as large as you want, but this has repercussions on the CPU time. We set $r = 500$ m taking as starting point the report by Konagai and colleagues about a landslide occurred in 2001 in El Salvador, that covered a runout distance of 500 m [30]. The estimate of the minimum extension of the radius (*r*) of the $HazardArea_i$ (Definition 4) is hard because of the different types of landslides (see, for example [31]) and the many physical and environmental variables that are involved in such events.

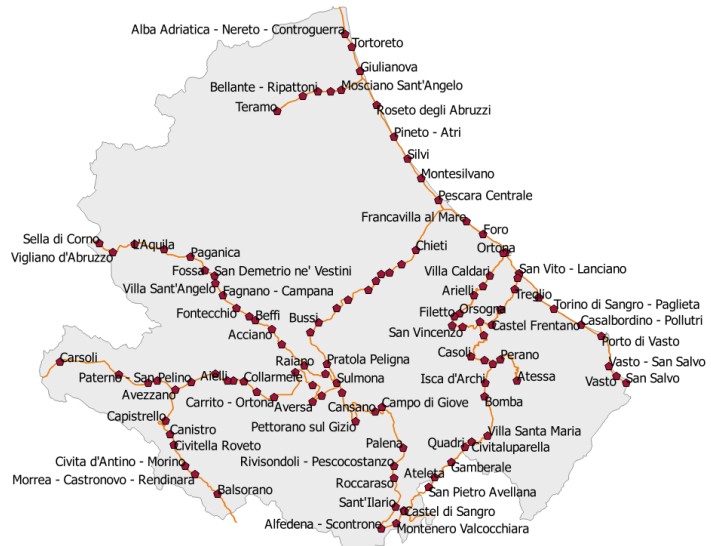

**Figure 8.** The railway stations along the railway lines of the Abruzzo region.

*3.2. The Results*

The map of Figure 9 shows the results obtained by executing the implemented PL/pgSQL code. Beside the symbol denoting each station, you can see the exposure value. Table 1 shows the ranking of the 114 railway stations of Abruzzo. For the sake of brevity, the stations are identified by the ID internal to the SpatialDB rather than by their name. The results returned by the proposed method are grouped in the exposure classes: High, Medium and Low, drawing inspiration from a widespread practice for risk classification studies (e.g., [32]). These three classes are matched, respectively, to the color red, yellow and green. The criteria used to group the 114 numeric values of Table 1 into the three classes is the following: the assets whose exposure is above 1.0 go to the High class, the assets whose exposure is below 0.4 go to the Low class, all the remaining assets go to the Medium class.

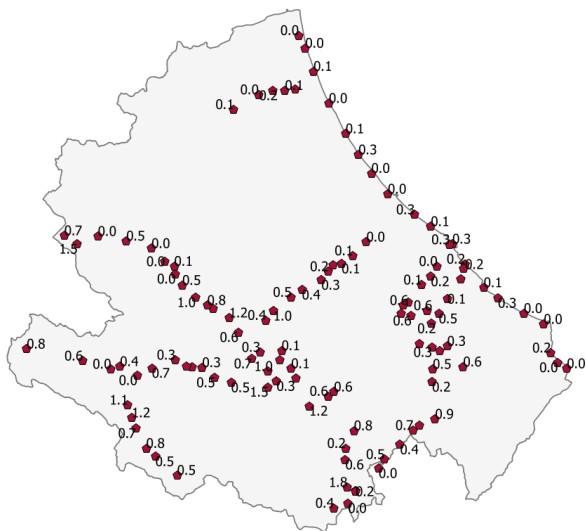

**Figure 9.** A pictorial representation of the ranking of the 114 railway stations of Abruzzo.

**Table 1.** The ranking of the 114 railway stations of Abruzzo.

| ID | Exposure | ID | Exposure | ID | Exposure | ID | Exposure |
|----|----------|----|----------|----|----------|----|----------|
| 44 | 1.75 | 78 | 0.56 | 6 | 0.30 | 108 | 0.08 |
| 65 | 1.52 | 97 | 0.56 | 15 | 0.29 | 28 | 0.07 |
| 62 | 1.51 | 88 | 0.54 | 11 | 0.29 | 23 | 0.07 |
| 114 | 1.24 | 68 | 0.54 | 98 | 0.28 | 103 | 0.07 |
| 82 | 1.17 | 94 | 0.53 | 73 | 0.27 | 37 | 0.07 |
| 38 | 1.15 | 102 | 0.53 | 47 | 0.26 | 25 | 0.05 |
| 81 | 1.06 | 60 | 0.52 | 30 | 0.26 | 21 | 0.04 |
| 66 | 1.04 | 85 | 0.51 | 104 | 0.25 | 106 | 0.02 |
| 54 | 1.01 | 69 | 0.51 | 29 | 0.25 | 46 | 0.02 |
| 33 | 0.98 | 72 | 0.50 | 100 | 0.22 | 17 | 0.01 |
| 92 | 0.90 | 55 | 0.49 | 22 | 0.20 | 16 | 0.01 |
| 91 | 0.89 | 86 | 0.48 | 18 | 0.19 | 113 | 0.01 |
| 41 | 0.83 | 32 | 0.48 | 79 | 0.19 | 56 | 0.01 |
| 84 | 0.81 | 45 | 0.45 | 13 | 0.19 | 2 | 0.00 |
| 80 | 0.78 | 111 | 0.44 | 71 | 0.18 | 77 | 0.00 |
| 53 | 0.76 | 89 | 0.43 | 48 | 0.18 | 75 | 0.00 |
| 74 | 0.73 | 76 | 0.42 | 107 | 0.17 | 1 | 0.00 |
| 67 | 0.71 | 34 | 0.42 | 93 | 0.16 | 87 | 0.00 |
| 52 | 0.70 | 105 | 0.41 | 42 | 0.16 | 24 | 0.00 |
| 83 | 0.68 | 109 | 0.39 | 36 | 0.14 | 8 | 0.00 |
| 90 | 0.67 | 31 | 0.37 | 35 | 0.14 | 61 | 0.00 |
| 63 | 0.65 | 95 | 0.35 | 26 | 0.13 | 7 | 0.00 |
| 43 | 0.64 | 50 | 0.34 | 27 | 0.12 | 59 | 0.00 |
| 39 | 0.63 | 12 | 0.33 | 20 | 0.12 | 58 | 0.00 |
| 110 | 0.60 | 70 | 0.32 | 5 | 0.11 | 49 | 0.00 |
| 51 | 0.59 | 99 | 0.32 | 10 | 0.11 | 4 | 0.00 |
| 40 | 0.58 | 96 | 0.32 | 57 | 0.11 | 19 | 0.00 |
| 112 | 0.57 | 9 | 0.31 | 3 | 0.09 | | |
| 101 | 0.57 | 64 | 0.31 | 14 | 0.08 | | |

The mapping from a ranking (e.g., about exposure values) to classes (e.g., High, Medium and Low) is a crucial issue whatever domain is about. In [33], for instance, a web survey is used to rank and classify finance journals by quality and importance. In defining the journal classes, Currie and Pandher adopt *thresholds* that are consistent with established practice and previous studies on journal assessment. Unfortunately, in our case no established thresholds are available, therefore any definition of the intervals is questionable. However, this does not nullify the value of the study whose main focus

is on the ranking itself. As mentioned in the Section 1, the availability of a ranking about strategic assets inside a GeoArea enables to overcome data intensive assessments on that scale. In fact, especially when the number of assets to be monitored is large, returning to the practitioners in risk mitigation the ranking of the assets they are in charge of guides them in prioritizing the controls on the field. In essence, they may limit the detailed risk assessment to the assets with a value of the exposure above a given threshold (i.e., the top-N assets). Two reasons motivate the mapping of ranking values into classes. One reason deals with the prodution of coloured exposure maps that are of great practical utility to the stakeholders involved in environmental protection programs (see, for instance [14]). The other reason relates to the validation of the adopted classification algorithm. Section 4 is about this point.

Among the stations, Sant'Ilario is characterized by the highest value of the exposure parameter (1.75). The three maps of Figure 10 give a visual explanation of such a result. In [34], Di Felice reports about the same case study, but carried out by applying the method in [19]. According to such a ranking, the station at the first position is Quadri, while Sant'Ilario is in the third position. The three maps of Figure 11 are about the Quadri station. The visual comparison of the maps in Figure 10 and those in Figure 11 proves that the new result is much more plausible than the previous one. The inconsistency arises because the method in [19] does not take into account the terrain elevation.

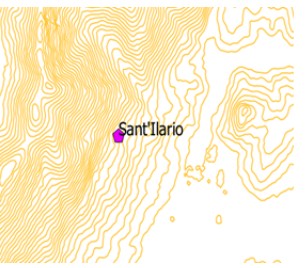 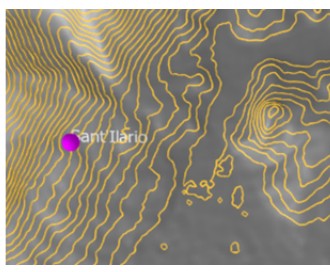 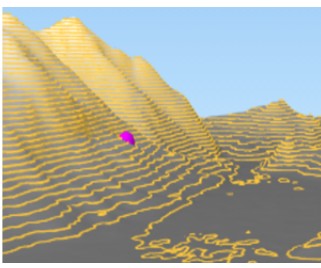

**Figure 10.** Three images of the terrain around the Sant'Ilario station. From left to right: a 2D map showing the contour lines near by the station; an up 3D map, and a frontal 3D map.

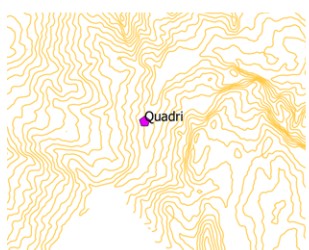 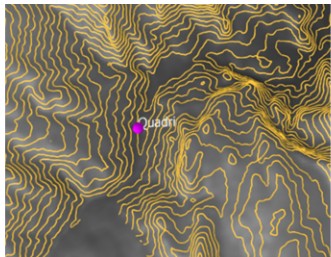 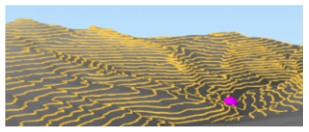

**Figure 11.** Three images of the terrain around the Quadri station. From left to right: a 2D map showing the contour lines near the station; an up 3D map, and a frontal 3D map.

The ranking returned by the proposed method provides support:

- To the analysis techniques aimed at assessing the level of vulnerability of assets;
- To the quickly identification of the buildings most exposed to the landslide hazard, among those located within the GeoArea;
- To set up a plan of actions aimed at the protection/evacuation of the buildings that are primarily exposed to the hazard because of a meteorological emergency alarm.

## 4. Validation of the Method for Ranking Assets Modeled as Points

In the Machine Learning domain the notion of "confusion matrix" (also called "contingency table") was introduced to assess the performance of a classification algorithm with respect to some test data. It is a two-dimensional matrix, indexed in one dimension by the true class of an object and in the other by the class that the classifier assigns [35].

The simplest way to validate the proposed ranking algorithm is to reduce the problem of the classification of experimental values to the case of binary contingency tables; that is, to the case in which only one class at a time is involved. Therefore, the validation problem can be formulated as follows.

Given n values ($v_1$, $v_2$, ..., $v_n$) and a class (C), construct the binary contingency table that summarizes how those values are classified both in the real world and as estimated by the algorithm of which we want to "measure" the effectiveness. Evidently, the value $v_1$ may fall in C or not, the same holds for $v_2$, ..., $v_n$.

The main diagonal of the contingency table has the number of elements predicted correctly. The total number of elements belonging to a real class is equal to the sum of the values on the corresponding row of the table. The total number of the elements present in the set involved in the classification operation is equal to the sum of all the totals.

Table 2 shows the structure of a generic binary contingency table. In the following the terms true positives, false positives, true negatives, and false negatives are defined (they are positive integers):

- True Positives (TP). This quantity denotes the cases that the classification algorithm has recognized correctly belonging to the class.
- False Positives (FP). This quantity denotes the cases of wrong classification. In practical terms, a false positive constitutes a false alarm.
- True Negatives (TN). This quantity denotes the cases that the algorithm has recognized correctly not belonging to the class.
- False Negatives (FN). This quantity denotes the cases for which the algorithm has confused the class to which an element belongs to. In the context of this article, these are cases of non-alarm. An error of enormous potential gravity.

**Table 2.** Structure of a binary contingency table.

|  |  | Predicted Class | | Total |
|---|---|---|---|---|
|  |  | **Class** | **No Class** |  |
| Actual Class | Class | TP | FN | P = TP + FN |
|  | No Class | FP | TN | N = FP + TN |
|  | Total | TP + FP | FN + TN | P + N |

*4.1. Validation Metrics*

Many metrics have been proposed to judge the goodness of an algorithm that reconstructs the observed reality (reference [36] is an authoritative source on the subject). The most common of them are listed below. The value of the metrics expresses a marginal probability, between 0 and 1.

The True Positive Rate (TPR, often called Recall) is defined as the percentage of positive cases correctly recognized as such (by the adopted classification method). In formulas:

$$TPR = \frac{TP}{TP + FN} \tag{8}$$

The True Negative Rate (TNR) denotes the percentage of actual negatives that are correctly identified as such. In formulas:

$$TNR = \frac{TN}{N} = \frac{TN}{FP + TN} \tag{9}$$

The Precision (P) is defined as:

$$P = \frac{TP}{TP + FP} \tag{10}$$

The Accuracy (Acc) is defined as:

$$Acc = \frac{TP + TN}{P + N} = \frac{TP + TN}{(TP + FN) + (TN + FP)} \tag{11}$$

*4.2. The Expected Ranking*

The 114 railway stations of the case study were classified, with respect to level of exposure to the landslide hazard, by exploiting the high degree of detail of the territory offered by Google. The buildings located on steep terrain were placed in the High class, in the Low class those located on flat land; while the Medium class was attributed to the hybrid situations (Table 3). In Table 3, the stations are identified by the ID internal to the SpatialDB. The match between the ID and the name is shown in Table 4 for the six stations that fall in the High class of exposure.

**Table 3.** The manual classification of the stations.

| Class | Number of Railway Stations | List of Their ID |
|---|---|---|
| High | 6 | 38, 44, 62, 65, 82, 114 |
| Medium | 36 | 32, 33, 39, 40, 41, 43, 51, 52, 53, 54, 55, 60, 63, 66, 67, 68, 69, 72, 74, 78, 80, 81, 83, 84, 85, 86, 88, 90, 91, 92, 94, 97, 101, 102, 110, 112 |
| Low | 72 | 1, 2, 3, 4, 5, 6, 7, 8, 9, 10, 11, 12, 13, 14, 15, 16, 17, 18, 19, 20, 21, 22, 23, 24, 25, 26, 27, 28, 29, 30, 31, 34, 35, 36, 37, 42, 45, 46, 47, 48, 49, 50, 56, 57, 58, 59, 61, 64, 70, 71, 73, 75, 76, 77, 79, 87, 89, 93, 95, 96, 98, 99, 100, 103, 104, 105, 106, 107, 108, 109, 111, 113 |

**Table 4.** The most exposed stations.

| ID | Name of the Railway Station |
|---|---|
| 44 | Sant'Ilario |
| 65 | Aversa |
| 62 | Vigliano d'Abruzzo |
| 114 | Acciano |
| 82 | Canistro |
| 38 | Pettorano sul Gizio |

The map of Figure 12 shows the 114 stations and their level of exposure (highlighted by the color) to the landslide hazard. Comparing this map with the knowledge of the elevation of the terrain of the Abruzzo region, there is a first confirmation of what was reasonable to expect, namely that the stations located on the part of the Region that faces the Adriatic sea do not run any landslide hazard and this because the ground is flat. Moreover, from this map we learn that the number of the most exposed stations is low (six out of 114) and, as it was predictadable, they are located in the hinterland of the Region that is mountainous and, therefore, subject to landslides during prolonged rains.

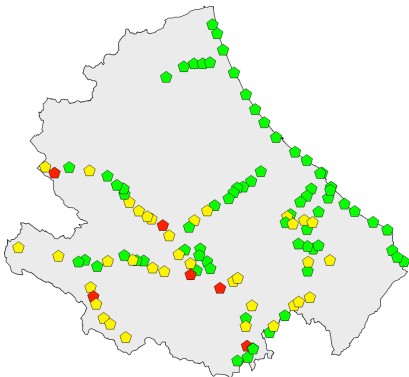

**Figure 12.** The railway stations of Abruzzo and their expected level of exposure to the landslide hazard.

*4.3. Evaluation of the Stations' Ranking*

The 3-class contingency table of Table 5 summarizes the result of the comparison between the manual ranking (Table 3) with that returned by the proposed classification algorithm (Table 1). Overall, 13 cases (out of 114) of classification mismatch arise from the comparison. In detail, from Table 5 it follows that the proposed algorithm:

- Classifies correctly 64 stations (out of 72) as belonging to the Low actual class, while it assigns the remaining eight to the Medium class;
- Classifies correctly 31 stations (out of 36) as belonging to the Medium actual class, while it puts two stations in the Low class and three in the High class;
- Classifies correctly the six stations belonging to the High actual class.

**Table 5.** The 3-class contingency table for the railway stations of Abruzzo.

|  |  | Predicted Classes | | | Total |
|---|---|---|---|---|---|
|  |  | Low | Medium | High |  |
|  | Low | 64 | 8 | 0 | 72 |
| Actual Classes | Medium | 2 | 31 | 3 | 36 |
|  | High | 0 | 0 | 6 | 6 |
|  | Total | 66 | 39 | 9 | 114 |

From Table 5 it is possible to derive the binary contingency Tables 6–8, and from them Table 9. The latter reports the quantitative assessment of the goodness of the results returned by the proposed ranking method as expressed by the metrics of Section 4.1.

**Table 6.** Binary contingency table for the class (of exposure) Low.

|  |  | Predicted Class | | Total |
|---|---|---|---|---|
|  |  | Low | No Low |  |
| Actual class | Low | 64 | 8 | 72 |
|  | No Low | 2 | 40 | 42 |
|  | Total | 66 | 48 | 114 |

**Table 7.** Binary contingency table for the class (of exposure) Medium.

|  |  | Predicted Class | | Total |
|---|---|---|---|---|
|  |  | Medium | No Medium |  |
| Actual Class | Medium | 31 | 5 | 36 |
|  | No Medium | 8 | 70 | 78 |
|  | Total | 39 | 75 | 114 |

**Table 8.** Binary contingency table for the class (of exposure) High.

| | | Predicted Class | | Total |
| | | High | No High | |
|---|---|---|---|---|
| Actual Class | High | 6 | 0 | 6 |
| | No High | 3 | 105 | 108 |
| Total | | 9 | 105 | 114 |

**Table 9.** Values of the metrics TPR, TNR, Precision and Accuracy for the Case Study.

| | Low | Medium | High |
|---|---|---|---|
| TPR | 0.889 | 0.861 | 1 |
| TNR | 0.952 | 0.897 | 0.972 |
| P | 0.970 | 0.795 | 0.667 |
| Acc | 0.912 | 0.886 | 0.974 |

The value of metric TPR is very good for the classes Low and Medium, optimal for the class High. The value of the TNR metric is also very good for the three classes. The Precision is very satisfactory for the class Low, while it decreases for the other two classes. The lowest value of the Precision ($p = 0.667$) occurs for the class High, although in that case the number of False Positives is not the highest of the three classes. This effect is determined by the fact that the most exposed stations are much less than those categorized in the Low and Medium classes (six vs. 72 and six vs. 36, respectively).

As usual, also in our proposal the result of the validation phase depends on *how* the numeric values are grouped into the classes. For example, let us suppose that the criteria for grouping the 114 numeric values of Table 1 are modified as follows: the assets whose exposure is above 1.10 go to the High class, the assets whose exposure is below 0.4 go to the Low class, all the remaining assets go to the Medium class. Table 10 shows how the 3-class contingency table (Table 5) is affected by the criteria. Now, 10 cases (out of 114) of classification mismatch arise. Most relevant is the fact that there are no more false positives about the assets of the High class. Table 11 shows the values of the four metrics based on the values in Table 10. From the comparison of the two pairs of tables (i.e., Table 10 vs. Table 5 and Table 11 vs. Table 9) it emerges that the second hypothesis of classification is better than the previous one. Unfortunately, given a generic case study, it is not possible to know in advance what would be the best classification criteria to be used. Our experience suggests to pay attention to the values of the interval about the High class. This is because it is a more desirable situation having false positives than false negatives for the assets in the top positions of the ranking, since a false positive means a false alarm, while a false negative means a missing alarm.

**Table 10.** An alternative 3-class contingency table for the railway stations of Abruzzo.

| | | Predicted Classes | | | Total |
| | | Low | Medium | High | |
|---|---|---|---|---|---|
| Actual Classes | Low | 64 | 8 | 0 | 72 |
| | Medium | 2 | 34 | 0 | 36 |
| | High | 0 | 0 | 6 | 6 |
| Total | | 66 | 42 | 6 | 114 |

**Table 11.** Values of the metrics TPR, TNR, Precision and Accuracy based on Table 10.

| | Low | Medium | High |
|---|---|---|---|
| TPR | 0.889 | 0.944 | 1 |
| TNR | 0.952 | 0.897 | 1 |
| P | 0.970 | 0.809 | 1 |
| Acc | 0.912 | 0.912 | 1 |

## 5. Materials and Methods for Assets Modeled as Lines

The formalization of the problem of computing the ranking of assets that can be modeled using the geometric primitive line and the proposal of a solving method constitute a consolidation of an article by Di Felice et al. [37]. The consolidation was achieved by taking into account the altimetric evolution of the GeoArea and extending, in an easy way, the ranking method formalized, implemented and validated in the first part of this paper. Preliminarily, it is necessary to introduce new definitions and notations.

### 5.1. Definitions and Notations

**Definition 18.** $\mathcal{R}$ (Routes)= $\{r_k(k = 1, ..., card(\mathcal{R})|r_k$ is a route that crosses the GeoArea}. Route is a generic line (e.g., a railway line, a highway, a power line, a pipeline, and so on) described by the tuple $\langle ID, name, geometry \rangle$. A route can be modeled as a "simple line", i.e., as a curve with two disconnected end-points, which does not pass in the same point more than once [21].

**Definition 19.** $\mathcal{RS}$ (Route Segments) = $\{rs_{k,s}(k = 1, ..., card(\mathcal{R})), (s = 1, ..., card(\mathcal{RS})) \mid rs_{k,s}$ is a route segment of $r_k$}. Route segment $rs_{k,s}$ is described by the tuple $\langle ID, route\_ID, geometry, exposure \rangle$.

**Definition 20.** $\mathcal{RSP}$ (Route Segment Points) = $\{rsp_{k,s,p}(k = 1, ..., card(\mathcal{R})), (s = 1, ..., card(\mathcal{RS})), (p = 1, ..., card(\mathcal{RSP})) \mid rsp_{k,s,p}$ is a route segment point of $rs_{k,s}$}. A generic $rsp_{k,s,p}$ is described by the tuple $\langle ID, segment\_ID, geometry, exposure \rangle$.

### 5.2. The Ranking Method

The method for computing the exposure values of the route segments of each route that crosses the GeoArea is composed of the three steps discussed below.

The first step (called Segmentation) "cuts" each route $r_k$ in segments of length $v$. The number of segments (denoted as $m_k$) is given by Equation (12); $m_k = card(\mathcal{RS})$.

$$m_k = \left\lceil \left( \frac{Length(r_k)}{v} \right) \right\rceil \tag{12}$$

Figure 13 exemplifies the "cutting" operation of a route. The segments have length $v$ with the exception of the last that might be shorter.

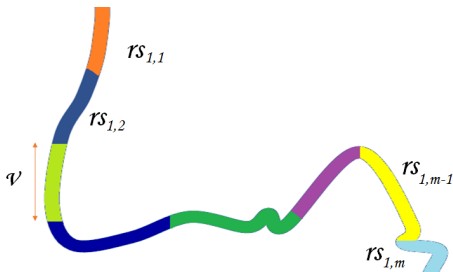

**Figure 13.** Segmentation of a route ($r_1$).

The second step (called Sampling) extracts points ($rsp_{k,s,p}$) from each segment $rs_{k,s}$; where $q$ (called the sampling step) denotes the maximum distance between two consecutive points of the same segment: $q << v$. The number of points returned by the sampling operation of segment $rs_{k,s}$ is given by Equation (13); $n_{k,s} = card(\mathcal{RSP})$.

$$n_{k,s} = \left\lceil \left( \frac{Length(rs_{k,s})}{q} \right) \right\rceil \tag{13}$$

Figure 14 exemplifies the "sampling" operation of a segment.

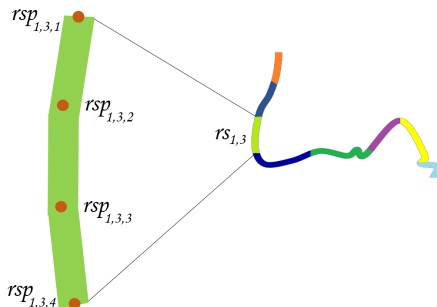

**Figure 14.** Sampling of a segment ($rs_{1,3}$).

The third and last step (called Calculation) associates an exposure value to each point extracted from the previous step and, then, computes the exposure of the various segments composing the routes.

The computation of the exposure for the points $rsp_{k,s,p}$ is carried out by applying the same method of Section 2.2.

Equation (14) computes the value of the exposure of the generic segment $rs_{k,s}$ of the route $r_k$. In the equation, $exp\_rsp_{k,s,p}$ denotes the value of the exposure of point $rsp_{k,s,p}$.

$$exp\_rs_{k,s} = \frac{1}{n_{k,s}} \left( \sum_{p=1}^{n_{k,s}} exp\_rsp_{k,s,p} \right) \tag{14}$$

### 5.3. Extension of the Spatial DataBase to the Routes

Below is shown the correspondence between the definitions in Section 5.1 and the names of the entities of the E-R schema that integrates that of Section 2.3.

- Routes $\leftrightarrow \mathcal{R}$;
- RouteSegments $\leftrightarrow \mathcal{RS}$;
- RoutePoints $\leftrightarrow \mathcal{RSP}$.

Tables below integrates the SpatialDB of Section 2.3.

- Routes(<u>id</u>, name, geom);
- RouteSegments(<u>id</u>, route_id, geom, exposure);
- RoutePoints(<u>id</u>, segment_id, geom, exposure).

## 6. Case Study: The Railway Lines of Abruzzo

The Abruzzo railway network consists of nine lines, for a total of about 685 km (Table 12).

**Table 12.** The railway lines of Abruzzo.

| Route Name | Extension (km) |
|---|---|
| Bologna–Bari | 124 |
| Ortona–Crocetta | 36 |
| Marina di San Vito–Castel di Sangro | 103 |
| Roma–Pescara | 170 |
| Avezzano–Roccasecca | 46 |
| Archi stazione–Atessa | 15 |
| Sulmona–Carpinone | 85 |
| Rieti–L'Aquila–Sulmona | 81 |
| Teramo–Giulianova | 25 |

The data about the Abruzzo railways were acquired from the Region homepage as shapefiles; then, they were validated, filtered, and stored into the PostgreSQL table `Routes`.

*6.1. The Ranking of the Railway Lines: Summary of the Results*

The nine railway lines were segmented into 460 stretches of 1500 m that, in turn, were sampled with a step of 300 m giving rise to 2740 points. For the points: (0 <= exposure <= 2.66), while for the segments: (0 <= exposure <= 2.43).

The maps of Figure 15 propose an overview of the results about the level of exposure to the landslide hazard (highlighted through the colors) of the Abruzzo railway lines. The three classes of exposure are the same of buildings (High, Medium and Low).

Because the method that ranks the routes makes use of the method for ranking the buildings, the latter already validated with satisfactory results (Section 4), it was not necessary to carry out its validation.

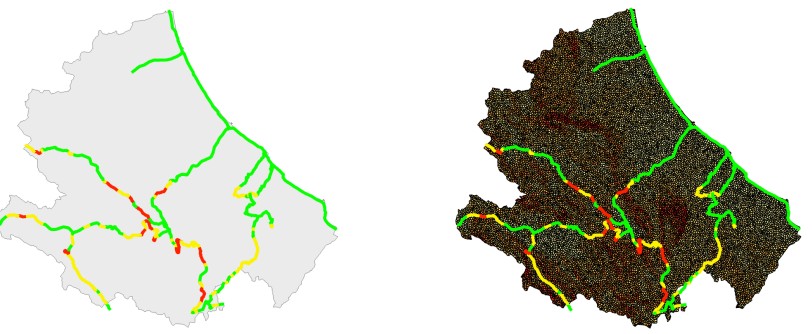

**Figure 15.** Two different views of the level of exposure of the railway lines of Abruzzo.

The results provide a confirmation of what was reasonable to expect, namely that the railway lines located along the part of the Region that faces the Adriatic sea do not run any landslide hazard and this because the ground is almost flat. Another finding, deriving from the examination of the maps of Figure 15, is that the number of the most exposed segments in the nine railway lines is small and, as was predictable, they are located in the hinterland of the Region that is mountainous and, therefore, subject to landslides in periods of prolonged rains.

Tables 13 and 14 provide, respectively, data about the distribution of the segments of the nine Abruzzo's railway lines and the corresponding sampled points, into the three exposure classes. These global data create an overall picture about the segments of the Region's network that are most exposed to the landslide hazard. The numbers inside Tables 13 and 14 were extracted by querying the SpatialDB.

**Table 13.** Distribution of the segments inside the three classes of exposure.

| Railway Line | Segment | High | Medium | Low |
|---|---|---|---|---|
| Archi stazione–Atessa | 10 | 0 | 1 | 9 |
| Avezzano–Roccasecca | 31 | 3 | 22 | 6 |
| Bologna–Bari | 83 | 0 | 0 | 83 |
| Marina di San Vito–Castel di Sangro | 71 | 0 | 24 | 47 |
| Ortona–Crocetta | 24 | 0 | 9 | 15 |
| Rieti–L'Aquila–Sulmona | 54 | 13 | 11 | 30 |
| Roma–Pescara | 113 | 14 | 32 | 67 |
| Sulmona–Carpinone | 57 | 17 | 22 | 18 |
| Teramo–Giulianova | 17 | 0 | 0 | 17 |
| | 460 | 47 | 121 | 292 |

**Table 14.** Distribution of the sampled points inside the three classes of exposure.

| Railway Line | Points | High | Medium | Low |
|---|---|---|---|---|
| Archi stazione–Atessa | 60 | 0 | 6 | 54 |
| Avezzano–Roccasecca | 184 | 14 | 134 | 36 |
| Bologna–Bari | 496 | 0 | 0 | 496 |
| Marina di San Vito–Castel di Sangro | 413 | 0 | 158 | 255 |
| Ortona–Crocetta | 143 | 0 | 50 | 93 |
| Rieti–L'Aquila–Sulmona | 324 | 71 | 81 | 172 |
| Roma–Pescara | 678 | 75 | 194 | 409 |
| Sulmona–Carpinone | 341 | 102 | 131 | 108 |
| Teramo–Giulianova | 101 | 0 | 0 | 101 |
| | 2740 | 262 | 754 | 1724 |

### 6.2. Ranking of Railway Lines: Detailed Analysis

Table 15 shows the four Abruzzo railway lines having segments with value of the exposure falling into the High class. The table contains, for each line, the relative length (in kilometers), the number of segments most exposed to the landslide hazard, the total length (in kilometers) of these segments and, finally, the percentage value of the length of the segments with respect to the entire length of the railway line they belong to. The lines Sulmona–Carpinone, Roma–Pescara and Rieti–L'Aquila–Sulmona contain about 94% of the most exposed segments. These lines cross the interior of the Abruzzo region where are present the greatest asperities of the terrain.

**Table 15.** Number and total length of the most exposed segments.

| Railway Linee | Line Length (km) | Number of Segments in the High Class | Segments in the High Class | Length % |
|---|---|---|---|---|
| Avezzano–Roccasecca | 45.91 | 3 | 4.50 | 9.8 |
| Rieti–L'Aquila–Sulmona | 80.89 | 13 | 19.50 | 24.1 |
| Roma–Pescara | 169.91 | 14 | 21.00 | 12.4 |
| Sulmona–Carpinone | 85.08 | 17 | 25.50 | 30 |

Below, insights for the Rieti–L'Aquila–Sulmona railway line are given. It contains the 13 most exposed segments (Table 15), of which one has the maximum value of the exposure parameter (2.43). From the joint examination of the two maps of Figure 16, we learn that the exposure peak is located in correspondence of the segment that goes from km 13.5 to km 15.

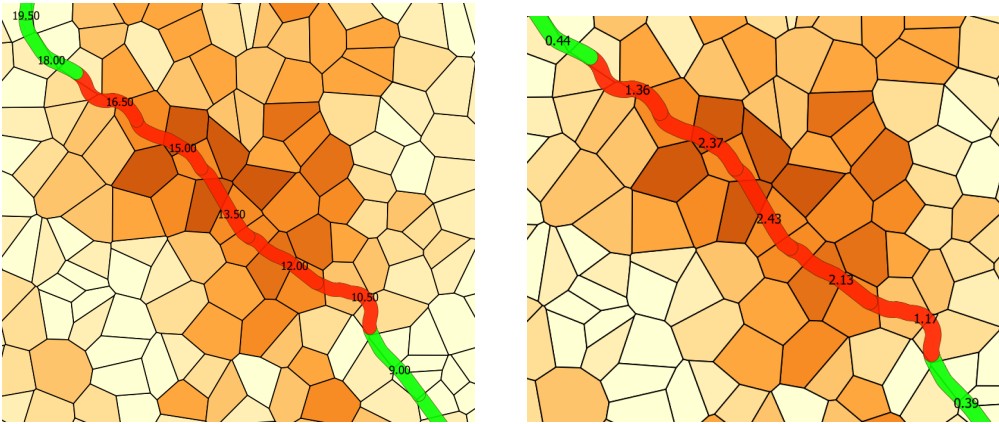

**Figure 16.** Few segments of the line Rieti–L'Aquila–Sulmona (**Left**) and the associated exposure values (**Right**).

By examining the altimetric trend of the Abruzzo's territory in correspondence of the most exposed segment (see the map of Figure 17), there is a further confirmation of the correctness of the diagnosis returned by the proposed method. In fact, they show that this segment of the line Rieti–L'Aquila–Sulmona crosses a gorge with two rather steep walls.

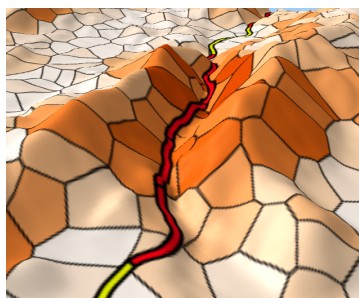

**Figure 17.** 3D rendering of the segments of Figure 16.

By querying the SpatialDB, it is also easy to knows the geographical coordinates of the end-points for each of the most exposed segments of the railway lines. Such data are fundamental when, in periods of prolonged rain in the GeoArea, it becomes necessary moving from the "control room" to the territory for inspecting the stability of the railway segments that are exposed to the highest level of landslide hazard.

## 7. Conclusions

The article's aim was the computation of the ranking of the assets present in a large territory with respect to their level of exposure to the landslide hazard. Assets that can be modeled as points and as lines were taken into account. For both the problems studied, an original method of resolution was proposed. The solution of the first problem is an important step for the solution of the second. Both the proposals constitute an improvement of what is already known in the literature on the subject.

The work began with the study of the state of the art, then it continued with the formalization of the two problems by means of equations, their transformation into algorithms, and their subsequent implementation. Then, the experimental phase was conducted at a geographical scale allowing a preliminary validation of the proposed methods. The ranking returned by the proposed methods is dramatically affected by the completeness and the quality of set $\mathcal{Z}$. In simple words, the ranking does not make much sense if, for the area of interest, there is not available a dataset $\mathcal{Z}$ built in terms of slope units as defined, for instance, in [26]. In the near future, we plan to carry out further case studies, as soon as other Italian regions will make available the shapefile implementing the $\mathcal{Z}$ dataset. The final goal is to make an accurate validation of the two prososed methods.

Our approach implements the adage "do more with less", well-known to public administrations which are experiencing budgetary contractions from many years. The list of the top-N assets most exposed to the landslide hazard appears the only viable way for these administrations to make the need for safety coexist with the containment of the time of controls and interruptions in service delivery and, therefore, with the costs to be paid.

Being able to use the results provided by methods such as those proposed in this paper is also of interest to private companies responsible for the safety at the national scale. For example, the Italian Railway Company operates a railway network of 16,734 km and more than 3000 stations located throughout the country (301,340 km$^2$). Implementing a periodic prevention business plan for each station and each kilometer of the railway network seems impractical.

Future work will focus on the integration of the GIS and the Wireless Sensor Network technology to set up a cooperating framework for the monitoring of the lines. In doing so, the ranking of the hotspots' level of landslide hazard will become dynamic, following environmental events in real-time.

**Funding:** This research was funded by a grant from the University of L'Aquila.

**Acknowledgments:** The manuscript has been improved thanks to pertinent criticisms from the referees. I'm infinitely grateful to them.

**Conflicts of Interest:** The author declares no conflict of interest.

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
