# Peer review of "Ranking of Assets with Respect to Their Exposure to the Landslide Hazard: A GIS Proposal"

_ijgi, doi:10.3390/ijgi9050326_

Round 1

Reviewer 1 Report

Overall the paper is well written and innovative from the scientific point of view. The arguments are well described and structured. The mathematical description of each variable is really valuable. I have appreciated the proposed method for exposure evaluation of the landslide vulnerable elements, but some comments and clarifications are necessary before acceptance. Please see the attached comments.

Author Response

The following references have been added to the revised manuscript:

Wang,F.; Xu,P.;Wang,C.;Wang, N.; and Jiang, N.

Application of a GIS-Based Slope UnitMethod for Landslide Susceptibility Mapping along the Longzi River, Southeastern Tibetan Plateau, China.

ISPRS Int. J. Geo-Inf., 2017, 6, 172; doi:10.3390/ijgi6060172

Eidsvig, U.M.K.; Kristensen, K.; and Vangelsten, B.V.

Assessing the risk posed by natural hazards to infrastructures.

Nat. Hazards Earth Syst. Sci., 17, 481–504, 2017, www.nat-hazards-earth-syst-sci.net/17/481/2017/

doi:10.5194/nhess-17-481-2017

Tobler, W.

A computer movie simulating urban growth in the detroit region.

Economic Geography, 46, 608 234–240 (1970)

==================== 

Overall the paper is well written and innovative from the scientific point of view.

The arguments are well described and structured.

The mathematical description of each variable is really valuable.

I have appreciated the proposed method for exposure evaluation of the landslide vulnerable elements, but some comments and clarifications are necessary before acceptance.

MAJOR COMMENTS:

  • Abstract - please explain the term ‘asset’ in Abstract. It is not clear from abstract itself

<<Done, see Line 1>>

  • Equation 1 – Please verify the equation. The general definition of Risk is

Hazard*Value OfElement At Risk*Exposure = Hazard*Vulnerability.

Otherwise, please explain better the components.

<<Eq.1 is as in Jaedicke et al., 2014; Ref. 25 in the paper. See Lines 38-41>>

  • The goal of the paper is to develop a ranking system of the exposure. In general the exposure is evaluated in relation to potential scenario regardless the probability of occurrence of the natural hazard considered (or in other words with probability equal to one). In this paper the exposure is evaluated considering a scenario defined by the trajectory of the potential landslide. As defined, exposure is not depending on the susceptibility, therefore, I cannot understand the relation between the landslide susceptibility and the product (Areai,j × IIlsi,j). Please explain.

<<The term Szk in Eq.6 allows to take into account the slop gradient. See Lines 113..121.>>

  • I think the terms Hazard, Susceptibility and Risk have been confused sometimes in the text. Hazard and susceptibility are referred to the physical event without relation with the elements at risk unless you are talking about risk (Hazard*Vulnerability).

But in this paper there are no discussions about the ValueOfElementAtRisk (Losses), thus the term risk should be used carefully (i.e Row 444-456).

Moreover, Hazard and Susceptibility are complementary but not synonymous.

Hazard = Susceptibility + assumption of landslide frequency (i.e. Tr 50 years), which requires a complete landslide inventory for the specific return period considered. In the text often the term Hazard have been used but the temporal reference (frequency) of the landslide has not been assumed. Please check.

<<I corrected the wrong occurences of the word risk. They are in blue in the paper. An example: Line 304>>

  • Row 289 – Why are you talking about hazard class? You are classifing exposure. Please explain.

<<AlI the wrong uses of the word HAZARD have been replaced with “exposure”. They are in blue in the paper. Few examples: Line 286, the captions of Tables 4, 6, 7 and 8>>

  • Row 256-279 – please add more information on the landslide inventory

<<Unfortunately I’m unable to say more, because nothing else is written in the metadata about the Z shapefile we got from the Environmental Agency of the Abruzzo region>>

  • Row 256-279 – Please report the use of the slope units as map units. This is a very important point because probably it is very difficult use the same method for pixel based susceptibility. I suppose it should be a basic assumption.

<<Good point. It has been implmented. See (new) Def.2, Lines 113..121>>

  • Row 256-279 – Please add the causative factors used

<<As I said above, it was impossible to say more.>>

MINOR COMMENTS

  • Row 55 – Please add reference of numbers

<<These numbers came by processing a shapefile about the Italian schools. So, there is no reference to be cited.>>

  • Row 57-65 – It is not clear. Please explain better.

<<See Lines 51..57>>

  • Equation 6 – Areai,j is not defined

<<See Eq.6 and Lines 216..218>>

  • Is the value of slope gradient taken in consideration in LS?

<<Yes, it is. The value of Szk embodies in itself info about the slop gradient. Greater is the slope angle, bigger is the value of Szk. That is why Szk is present in Eq.6. See Line 118>>

  • Def 18 - Please add image as example of partial_exposure

<<Ignored because another reviewer said that the document has to be reduced of 40%. My apologies.>>

  • Row 417 – please add image about the relation point-segment

<<See Fig.14>>

Reviewer 2 Report

The aim  of this interesting article is to rank assets in a large territory w.r.t. exposure to landslide hazards, and to classify their exposure as high, medium or low. The ranking variable is the exposure level, and is defined in this article. My main concern is why this exposure level should be meaningful, as there is not much motivation as why this particular function was chosen for buildings, resp. railway lines. For the classification, it is not clear to me how the particular mapping from exposure levels to the classes is motivated. These issues are crucial, if the method is to be useful for general geographic data. For these reasons, I suggest a major revision.

A detailed remark:

Please separate figure, table or equation numbers from the words 'Fig.', 'Tab.' or 'Eq.' as in 'Fig. 2' instead of 'Fig.2'.

Author Response

The following references have been added to the revised manuscript:

Wang,F.; Xu,P.;Wang,C.;Wang, N.; and Jiang, N.

Application of a GIS-Based Slope UnitMethod for Landslide Susceptibility Mapping along the Longzi River, Southeastern Tibetan Plateau, China.

ISPRS Int. J. Geo-Inf., 2017, 6, 172; doi:10.3390/ijgi6060172

Eidsvig, U.M.K.; Kristensen, K.; and Vangelsten, B.V.

Assessing the risk posed by natural hazards to infrastructures.

Nat. Hazards Earth Syst. Sci., 17, 481–504, 2017, www.nat-hazards-earth-syst-sci.net/17/481/2017/

doi:10.5194/nhess-17-481-2017

Tobler, W.

A computer movie simulating urban growth in the detroit region.

Economic Geography, 46, 608 234–240 (1970)

====================== 

The aim  of this interesting article is to rank assets in a large territory w.r.t. exposure to landslide hazards, and to classify their exposure as high, medium or low.

The ranking variable is the exposure level, and is defined in this article.

My main concern is why this exposure level should be meaningful, as there is not much motivation as why this particular function was chosen for buildings, resp. railway lines.

<<Several parts of the original manuscript have been rewritten. New sentences were added, too. See sentences in blue.>>

For the classification, it is not clear to me how the particular mapping from exposure levels to the classes is motivated. These issues are crucial, if the method is to be useful for general geographic data.

<<In the submitted manuscript the terms exposure, hazard and risk have been confused, sometimes. This flaw has been fixed. (See the sentences in blue.) Now, the motivation should be clearer.>>

A detailed remark:

Please separate figure, table or equation numbers from the words 'Fig.', 'Tab.' or 'Eq.' as in 'Fig. 2' instead of 'Fig.2'.

<<All the shortenings (Fig., Tab., Def., Eq.) have been expanded.>>

Reviewer 3 Report

It is an interesting subject and a well written article. However, in its current format it is too long and it is more in a report format. I, therefore, suggest shortening it to a maximum of 6000 words. For example, some of the technical details can be cut out, present as appendix or shorten.   

I would also suggest a discussion section in which the results are compared with other approaches and studies (even the author's similar publications). The section should also discuss the other existing methods of risk analysis and assessment. This should include elements of uncertainty associated with data, and data collection (e.g. geotechnical parameters of soil) and data collection. 

Also further improvement of the developed model should be discussed such as utilising advance machine learning techniques or benefiting from live data and even potential development of a digital twin could be discussed. 

Author Response

The following references have been added to the revised manuscript:

Wang,F.; Xu,P.;Wang,C.;Wang, N.; and Jiang, N.

Application of a GIS-Based Slope UnitMethod for Landslide Susceptibility Mapping along the Longzi River, Southeastern Tibetan Plateau, China.

ISPRS Int. J. Geo-Inf., 2017, 6, 172; doi:10.3390/ijgi6060172

Eidsvig, U.M.K.; Kristensen, K.; and Vangelsten, B.V.

Assessing the risk posed by natural hazards to infrastructures.

Nat. Hazards Earth Syst. Sci., 17, 481–504, 2017, www.nat-hazards-earth-syst-sci.net/17/481/2017/

doi:10.5194/nhess-17-481-2017

Tobler, W.

A computer movie simulating urban growth in the detroit region.

Economic Geography, 46, 608 234–240 (1970)

====================================== 

It is an interesting subject and a well written article.

However, in its current format it is too long and it is more in a report format. I, therefore, suggest shortening it to a maximum of 6000 words.

For example, some of the technical details can be cut out, present as appendix or shorten.   

<<The other 3 reviewers didn’t say a word against the paper’s length, so this comment has not been implemented. However, I did my best to simplify sentences, especially in Sec.1. Moreover Figg. 1, 2, 9, 11, 20, 21, and 22 have been removed. A new one was  added because required by Rev.#1.>>

I would also suggest a discussion section in which the results are compared with other approaches and studies (even the author's similar publications).

<<As written in the revised manuscript (Lines 74..75) the formalization of the two novel methods is the actual contribution of the present paper, while the Case study is devoted to carry out a preliminary validation of the proposed methods. That’s why a more in-deep discussion of the results was not recommended by the other three reviewers.

In the paper are present sentences that link the present work with previous research.>>

The section should also discuss the other existing methods of risk analysis and assessment.

<<As correctly written by Rev.#1: “The goal of the paper is to develop a ranking system of the exposure”. That’s why existing methods of risk analysis and assessment are just touched in Sec.1>>

This should include elements of uncertainty associated with data, and data collection (e.g. geotechnical parameters of soil) and data collection. 

<<Lines 484..489 have been added.>>

Also further improvement of the developed model should be discussed such as utilising advance machine learning techniques or benefiting from live data and even potential development of a digital twin could be discussed. 

<<I fully agree with the reviewer about the relevance of “live data” to keep the ranking up-to-date over time.

The last paragraph of the manuscript talks about our plan of making recourse to WSNs for the future work>>

Reviewer 4 Report

The article is interesting. The results of the research can be useful for a wider range of users.

The introduction is overloaded with information about the risks which don’t contribute much to the article.

Please explain clearly what means, e.g. Z3.3 . Does the figure show several GeoArea? Why Zones number doesn’t start with the first zone? Is it a deliberate action? How the zones were selected for GeoArea?

What algorithm was used to select zone boundaries? Why this algorithm was chosen? has it been proven and described in some article that this one is the best? Are there other algorithms to choose right zones ? What were the criterias for selecting the algorithm? The statement that the program generates this in such a way is not convincing enought.

The lines 128. How the centre of the building was determined? Has it derived from the mathematical formula? For buildings which are spatially diverse their centroids are not always located "on the buildings".

The lines 132 Radius r is too simplistic. The tectonic faults, geological lines and other they don't have that character. It should be assumed for testing assumptions only. The practical results can be different. There is also no information about land gradients in the first part of the article. This is crucial for landslide risk area modelling.

Figure 7 What does green mean?

Figure 18 Is on this figure the whole line Rieti- L'Aquila - Sulmon, or only fragments of it.

Figure 19. Were the zones stretched due to the gradient?

The figures 20 and 21 are unnecessary. The author should choose figure 19 or 20.

The chapter 2.1 is not readable. The simple things are described in a inaccessible way.

The signatures under the figures are insufficient. They don’t indicate what the figures shows. They're tiring for the reader with lots of abbreviations, markings and definitions.

The introduction suggests that buildings will be used as schools but as a result there are only railway stations. There are other factors which can be used on the urbanised areas as acting landslide barriers

The article is compatible with the theme of the journal but is incomprehensible and intricate in some parts.The main purpose of the article is lost.

Case Study: the Railway Stations of Abruzzo is appropriate. It clearly shows how the algorithm works.

Does the table 4 bring something relevant to the article? The same table 3? They are unnecessary.

I suggest to replace Table 1 with a graph to shorten an article that is too long.

Author Response

The following references have been added to the revised manuscript:

Wang,F.; Xu,P.;Wang,C.;Wang, N.; and Jiang, N.

Application of a GIS-Based Slope UnitMethod for Landslide Susceptibility Mapping along the Longzi River, Southeastern Tibetan Plateau, China.

ISPRS Int. J. Geo-Inf., 2017, 6, 172; doi:10.3390/ijgi6060172

Eidsvig, U.M.K.; Kristensen, K.; and Vangelsten, B.V.

Assessing the risk posed by natural hazards to infrastructures.

Nat. Hazards Earth Syst. Sci., 17, 481–504, 2017, www.nat-hazards-earth-syst-sci.net/17/481/2017/

doi:10.5194/nhess-17-481-2017

Tobler, W.

A computer movie simulating urban growth in the detroit region.

Economic Geography, 46, 608 234–240 (1970)

========================================= 

The article is interesting. The results of the research can be useful for a wider range of users.

The introduction is overloaded with information about the risks which don’t contribute much to the article.

<<I did my best to shrink Sec. 1.>>

Please explain clearly what means, e.g. Z3.3. 

Does the figure show several GeoArea?

Why Zones number doesn’t start with the first zone? Is it a deliberate action?  

<<The previous 3 comments have been taken into account by removing Fig.1. It was tricky.>>  

How the zones were selected for GeoArea?  What algorithm was used to select zone boundaries?  Why this algorithm was chosen? Has it been proven and described in some article that this one is the best? Are there other algorithms to choose right zones ? What were the criterias for selecting the algorithm? The statement that the program generates this in such a way is not convincing enought.    

<<Lines 113..121 answers the questions above.>>

The lines 128. How the centre of the building was determined? Has it derived from the mathematical formula? For buildings which are spatially diverse their centroids are not always located "on the buildings".  

<<Thanks to this comment, Def. 3 was generalized by replacing the point position of buildings with their actual geometry (not always available, as in our Case Study). In turn, Def. 4 (BuildingBuffer_i) was cancelled because useless.>> 

The lines 132 Radius r is too simplistic.

The tectonic faults, geological lines and other they don't have that character. It should be assumed for testing assumptions only. The practical results can be different.

<<See paper, the new Def.4 is general.>>

There is also no information about land gradients in the first part of the article. This is crucial for landslide risk area modelling.

<<The new version of the manuscript is explicit about this point. See Def.2, Lines 113..121>>

Figure 7 What does green mean?

<<See Def. 15 (Lines 177..178)>>

Figure 18 Is on this figure the whole line Rieti- L'Aquila - Sulmon, or only fragments of it.

<<The Figure shows a portion of the whole line. See caption of Fig.16>>

Figure 19. Were the zones stretched due to the gradient?

<<Yes. See Def.2>>

The figures 20 and 21 are unnecessary. The author should choose figure 19 or 20.

<<Fig. 20 and Fig. 21 have been removed>>

The chapter 2.1 is not readable. The simple things are described in a inaccessible way.

<<I understand this point of view. Readability suffers because of the formalization of the basic concepts of the proposed methods.

Rev.#1’s opinion is different:

Overall the paper is well written and innovative from the scientific point of view.

The arguments are well described and structured.

The mathematical description of each variable is really valuable

As that of Rev.#3:

It is an interesting subject and a well written article.”>>

The signatures under the figures are insufficient.

They don’t indicate what the figures shows.

They're tiring for the reader with lots of abbreviations, markings and definitions.

<<Most captions were rewritten.>>

The introduction suggests that buildings will be used as schools but as a result there are only railway stations.

<<Done. See Lines 62 and 186>>

There are other factors which can be used on the urbanised areas as acting landslide barriers.

<<Out of scope.>>

The article is compatible with the theme of the journal but is incomprehensible and intricate in some parts. The main purpose of the article is lost.

<<I hope that the many corrections and integrations made to the original manuscript have enhanced its readability.>>

Case Study: the Railway Stations of Abruzzo is appropriate. It clearly shows how the algorithm works.

Does the table 4 bring something relevant to the article? The same table 3? They are unnecessary.

<<Tabb. 3 and 4 are still there because the other 3 reviewers didn’t say a word against them. My apologies.>> 

Round 2

Reviewer 1 Report

Well, more or less the previous comments have been changed and considered. Last comment, I suggest to check again the use of the terms: risk, hazard and exposure (i.e. risk in row 456?)

Author Response

Last comment, I suggest to check again the use of the terms: risk, hazard and exposure (i.e. risk in rowc456?)

<<The mistake has been removed (Line 474). Thanks.>>

Reviewer 2 Report

The author has addressed my issues raised in the first round: motivation of the variable "exposition level", and the classification of its range into three classes High, Medium and Low.

The exposure level is now sufficiently motivated. However, the issue about the motivation for the classes High, Medium and Low are not duly addressed: Which intervals do High, Medium and Low precisely mean? And why are these intervals defined in that particular way? The issue is that of producing three clusters from a given exposure range (between 0 and 2.66, or between 0 and 2.43, respectively). What are the cluster definitions, and why are they meaningful?

The extended length of the article is appropriate, because many new notions need to be explained at such a detail in order for the reader to understand them.

My recommendation is another major revision.

Detailed remarks
- l. 53. "guide" -> "guides"
- l. 118. "Greater is ..." -> "The greater the slope gradient is, the bigger is ..."
- l. 128. After "Hazard Area_i" put a dot.
- l. 133. "In abstract" -> "An abstract"
- Figure 1. "fucsia" -> "fuchsia"
- l. 486. "it is not available" -> "there is not available"
- l. 487. "In the next future" -> "In the near future"

Author Response

The issue about the motivation for the classes High, Medium and Low are not duly addressed:

Which intervals do High, Medium and Low precisely mean? And why are these intervals defined in that particular way? The issue is that of producing three clusters from a given exposure range (between 0 and 2.66, or between 0 and 2.43, respectively). What are the cluster definitions, and why are they meaningful?

<<Lines 294..309 have been added, together with two further references:

[Currie and Pandher, 2020] and [Promper and Glade,2016].

Lines 387..400 are part of the reply to your proper comment.>>

Detailed remarks

<<All the detailed remarks have been implemented (see paper). Thanks a lot.>>

Reviewer 4 Report

I accept.

Author Response

Thanks

Round 3

Reviewer 2 Report

My concerns are now sufficiently addressed. I recommend acceptance.

Detailed remark:

Line 398. "This because, it is ..." -> "This is because it is ..."

Author Response

Once again tanks for your kindness and patience.